# Fixation with Carbon Fiber Plates After Curettage in Benign and Locally Aggressive Bone Tumors: Clinical and Radiographic Outcomes

**DOI:** 10.3390/jcm14072371

**Published:** 2025-03-29

**Authors:** Edoardo Ipponi, Elena Bechini, Vittoria Bettarini, Martina Cordoni, Fabrizia Gentili, Antonio D’Arienzo, Paolo Domenico Parchi, Lorenzo Andreani

**Affiliations:** Department of Orthopedics and Trauma Surgery, University of Pisa, Via Paradisa 2, 56124 Pisa, Italy

**Keywords:** PEEK, orthopedics, oncology, limb sparing, complications, recurrence, functionality

## Abstract

**Background**: Curettage represents a reliable therapeutic option for large-sized benign and locally aggressive bone tumors. In cases of impending fractures, internal fixation with plates and screws can be necessary to stabilize the treated bone after curettage. Metal plates have been the only fixation devices available on the market for decades, but Carbon-fiber-reinforced polyetheretherketone (CFR-PEEK) now represents an alternative in orthopedic oncology. **Methods**: We reviewed our patients with benign or locally aggressive bone tumors treated with curettage and fixation with CFR-PEEK plates. Plate length and curettage technique were chosen considering the characteristics of each lesion. We recorded the size and location of the lesions, adjuvant treatments and fillers used after curettage, complications, and local recurrences. Postoperative functionality was assessed using the MSTS score. **Results**: Forty cases were included in our study. The tumors were located in the distal femur (19 cases), femur shaft (1), humerus (17), or proximal tibia (3). Local adjuvants were used in 20 cases. Cavities were filled with bone allografts in 30 cases and cement in 10 cases. Only four cases suffered postoperative complications, and two developed local recurrences. The mean postoperative follow-up was 29.2 months. The mean postoperative upper and lower limb MSTS was 28.0 and 26.7, respectively. **Conclusions**: After an accurate curettage and an adequate filling of the resulting bone gap, CFR-PEEK plates can provide good mechanical resistance, and their radio-transparency can ease the early diagnosis of local recurrences. CFR-PEEK plates should be considered in selected cases, in a personalized surgical approach.

## 1. Introduction

Benign tumors of the bone consist of a wide variety of neoplasms developed in the skeleton. Their incidence is debated due to their often asymptomatic presentation and difficulty in detection, but it is generally accepted that benign conditions are 100 times more frequent than primary malignant bone tumors [1,2]. The therapeutic approach to benign tumors of the bone can vary depending on the behavior of each case. A correct evaluation of clinical presentations and imaging evidence is necessary to orientate the diagnosis correctly [3,4]. A biopsy is fundamental to define a histological diagnosis, especially for large lesions with rapid growth and radiological suspicion of local aggressiveness [5]. Surgery represents the treatment of choice for locally aggressive lesions and benign tumors whose growth could undermine the resistance of the host bone [6]. Curettage, in particular, represents an established and reliable surgical treatment for the treatment of nonmalignant primary bone tumors, consisting of the exposition of the neoplasm and its consequential removal in pieces using spoons and burrs until reaching macroscopically healthy bone tissue [7]. After curettage, the surfaces can receive further adjuvant treatments to bonify the area and minimize the risk of local recurrence [8,9]. The resulting cavity can then be filled with bone allografts, autografts, bone substitutes, or cement [9,10]. Depending on the size of the tumor, the cortical ballooning or scalloping it exerted, and the filler used, some treated bones can be so fragile that they require prophylactic stabilization and prevent postoperative pathologic fractures. For decades, orthopedic oncologists have relied on metallic plates and other metallic fixation devices to stabilize impending fractures and significant defects after curettage [11,12,13,14,15,16,17,18,19,20,21,22,23]. However, despite their resistance and relatively low price, metal plates in orthopedic oncology may have some drawbacks, such as a lower elasticity compared to the bone and the generation of imaging artifacts in CT and MRI scans that are often necessary to detect recurrences during the follow-up. These limitations could be overcome with the recent introduction of Carbon-fiber-reinforced polyetheretherketone (CFR-PEEK) implants in orthopedic oncology [24]. CFR-PEEK implants have gained increasing interest due to their biocompatibility and peculiar imaging and mechanical properties. Carbon’s radiolucency eases the visualization of bone healing and the early detection of eventual tumor recurrences. The absence of metallic artifacts would also enable more precise and reliable postoperative radiotherapy planning for metastatic lesions. From a mechanical point of view, carbon fiber’s elasticity and reduced stress shielding might be responsible for better quality of the healing bone [24]. This potential also comes with complication rates comparable to the ones recorded using metal plates, as reported in a literature review published by Rijs et al. in 2023 [24].

For these reasons, carbon fiber plates could theoretically represent a promising device for stabilizing bone segments treated with curettage and cement filling or bone grafting. However, modern literature still lacks a large case series on this topic [25,26]. 

In this study, we reported our experience using CFR-PEEK plates to stabilize bones treated with curettage due to benign or locally aggressive bone tumors.

## 2. Materials and Methods

This single-center retrospective study was conducted according to the ethical standards in the 1964 Declaration of Helsinki and its later amendments.

Our study reviewed all patients with benign or locally aggressive bone tumors treated with curettage and internal fixation with carbon fiber plates in our institution between July 2016 and February 2024. We collected data regarding each patient’s age and gender. Each patient had preoperative X-rays and MRIs to assess the localization of the disease, estimate its larger diameter, orient the diagnosis, and guide a personalized surgical planning for each case. Cases with massive preoperative scalloping, cortical ballooning, and clinical and radiographic signs suggestive of impending fractures were included in our study. Impending fractures were identified according to the Mirels’ criteria (score nine or higher) [27]. An established diagnosis of pathological fractures before surgery represented an exclusion criterion for our study. 

Surgical procedures were customized to the necessities of each single case. Intraoperatively, the cortical bone that covers the lesion was focally exposed, and a cortical window was shaped using an oscillating saw. Gently removing the cortical window, surgeons could expose the tumor. The intramedullary lesion was then removed with a vigorous curettage using a Volkmann curette and later a high-speed burr until reaching macroscopically healthy bone tissue. Our pathologists examined surgical specimens to formulate or confirm the histological diagnosis using routine histology, histochemistry, and immunohistochemistry techniques. 

For selected cases with massive scalloping or a known diagnosis of a locally aggressive tumor, the surfaces of the resulting cavities were eventually treated with local adjuvants, including alcohol, phenol, or cryotherapy with cryoprobes (Figure 1). 

The intramedullary bone cavity was later filled with a morselized bone allograft or bone cement (polymethyl methacrylate; PMMA), depending on the localization and the necessities of each case. Morselized bone allograft was preferred in younger patients, whereas cement was considered for elderly patients and lesions reaching the subchondral region.

Once the bone was filled, the cortical window was put back to restore the anatomical continuity of the bone surface or completely removed in case of massive scalloping and thinning. Carbon-fiber-reinforced polyetheretherketone (CFR-PEEK) buttress plates were used to cover the window area and stabilize the bone with compression and angular stable screws (Piccolo Composite™ Plating System, CarboFix Orthopedics^®^, Herzliya, Israel) (Figure 2). CarboFix nails and plates are made of continuous carbon fibers reinforced with polyether ether ketone (PEEK), which adheres the carbon fibers together. The continuous carbon fibers are layered in longitudinal and helical diagonal orientations.

Intraoperative fluoroscopy confirmed the complete removal of the neoplastic mass, the adequate filling with the augment of choice, and the correct positioning of the fixation devices. An X-ray was also taken at the end of each surgical procedure (Figure 3). 

The postoperative follow-up consisted of serial clinical evaluations, and postoperative MRIs were performed within one month (clinical evaluation only), 6 and 12 months after surgery, and later once per year for up to five years. Further or additional evaluations were scheduled depending on the necessities of each single case. Imaging evidence was used to diagnose local recurrences and assess the grade of integration according to the modified Neer classification for those who had been treated with bone grafting. 

Each complication with a grade II or higher, according to the Clavien–Dindo Classification, was reported. Each patient’s functional outcome of the treated limb was assessed at their latest follow-up using the upper and lower limb scoring systems of the Musculoskeletal Tumor Society (MSTS). Statistical analysis was performed using Stata SE 13 (StataCorp LLC, College Station, TX, USA). Statistical significance was set at 0.05 for all endpoints. 

## 3. Results

Forty patients met our inclusion criteria and were included in our study. Our cohort included 22 women and 18 men, with a mean age at surgery of 44.5 (14–74). The femur was the most involved bone, with 20 cases: 19 were located in the distal segment of the bone, and one was in the proximal epiphysis. Seventeen patients had their tumors localized in their humerus: 16 lesions were located in the proximal humerus, whereas only one was confined to the humeral shaft. The proximal tibia hosted the lesions of the remaining three cases. The mean size (larger diameter) of treated lesions was 7.4 (3–16). 

Twenty-eight patients were diagnosed with chondral lesions: 11 chondromas and 17 atypical chondromatous tumors (ACTs) (previously grade I chondrosarcomas). The remaining patients had a diagnosis of aneurysmal bone cyst (7 cases), giant cell tumor of the bone (GCT) (3), fibrous dysplasia (1), or Liposclerosing myxofibrous tumor (1).

Twenty cases did not receive any local adjuvant treatment, whereas many were treated with local cryotherapy (9), phenol (9), or alcohol (2). The cavity made during the curettage was filled with bone allografts in 30 cases and PMMA in the remaining 10. None of our cases had major intraoperative complications.

The mean postoperative follow-up was 29.2 months (10–101). At their latest follow-up, all 30 cases whose bone gap was filled with bone grafts had complete radiological healing without residual lytic areas (Modified Neer class I). 

One case developed a large aseptic seroma (10 cm wide) in his distal thigh, close to the surgical incision site, which required surgical incision and drainage. One patient, a 17-year-old young man with a grade I chondrosarcoma of the distal femur treated with curettage, cryotherapy, and allograft filling, was diagnosed with a deep infection within two months after surgery. The infection was eradicated with intravenous antibiotics, surgical debridement, and removal of the internal fixation devices (Figure 4).

Another patient with an atypical chondroma of the distal femur, treated with alcoholization and bone grafting, had a superficial infection that was successfully treated with oral antibiotic treatment. 

The case with a lesion in the femur shaft—a 68-year-old woman treated with curettage alone, cement filling, and plate fixation—fell and had direct trauma on her hip 10 months after the treatment. X-rays evidenced a displaced fracture of the ipsilateral femoral neck that required a total hip arthroplasty (THA). 

The remaining 36 cases (90%) had no major postoperative complications (Figure 5).

Two cases had local recurrences. One had an ACT (9.5 cm wide at the moment of surgery), and the other had a GCT of the bone (5.8 cm). The recurrences were diagnosed within 3 and 6 months after surgery, respectively. Both lesions were localized in the distal femur and had been treated with curettage, cryotherapy, cement filling, and stabilization with a lateral plate. The patient with a previous diagnosis of ACT (9.5 cm wide at the moment of surgery) had come back to good functional outcomes and was painless at our first outpatient visit. Still, the MRI scans performed within three months after surgery highlighted an endomedullary osteolytic area, continuous to the site treated with curettage. A needle biopsy revealed a local recurrence with focal histological degeneration to a grade 2 chondrosarcoma. The patient was therefore treated with place removal and a massive bone resection and replacement of the distal femur with a megaprosthetic implant. He still had a good functional recovery, with satisfying postoperative functional outcomes (MSTS 28 out of 30). Histological investigations on the surgical specimen confirmed a grade II chondrosarcoma evolution. The patient with a GCT, although silent from a clinical point of view, showed MRI signs of local recurrence. Three months after surgery, she developed a shaded area surrounding the bone cement used to fill the cavity after curettage. Three months later, at a similar examination, the area appeared like a clear nodularity, and a CT-guided needle biopsy confirmed the diagnosis of GCT local recurrence. It has been proposed to the patient to remove the plate and replace the distal femur with a megaprosthesis, but she is still considering her consent to perform further surgeries. The remaining 38 cases did not suffer from local recurrences through their postoperative intercourse.

At their latest follow-up, patients treated in their upper limbs had an MSTS score of 28.0 (22–30). The mean MSTS score of those treated in their lower limbs was 26.7 (18–30).

A summary of our patients’ results is reported in Table 1.

In our cohort, the filling method did not influence the functional outcome of our patients (Whitney–Mann test *p* = 0.871).

No statistically significant linear correlation between the tumor size and the postoperative MSTS score was detected (Pearson correlation test *p* = 0.901).

## 4. Discussion

Intralesional curettage represents the treatment of choice for benign or locally aggressive tumors. When performing curettage, surgeons should maintain the stability and structural integrity of the treated bone [7,8,9,10]. Avoiding unnecessary removal of healthy bone tissue preserves the quantity and quality of local bone stock, reducing the risk of intraoperative and postoperative damages. In some cases, however, the curettage significantly undermines the stability of the bone, despite the efforts and precautions of orthopedic surgeons [6]. 

Several authors have suggested that the size of treated lesions impacts the risk of postoperative fracture or deformity after surgical treatment [17,20,28,29]. Hirn et al. [28] showed a strong correlation between lesions’ volume and size and their risk of postoperative fracture. In their population of 146 cases, the mean lesion volume in those who later developed postoperative fractures was significantly wider (108 cm^3^) than those who did not experience such complications (58 cm^3^) (*p* = 0.003). In the same study, the global fracture risk was 3% in patients with lesions shorter than 5 cm, while it increased to 15% for those larger than 5 cm (*p* = 0.02). Toğral et al. [20] also showed that benign tumors or tumor-like lesions of the long bones had a higher fracture risk if their volume was greater than 67 cm^3^, especially if the patient’s age was over 35 years. 

Internal fixation with plates and screws can be used to counterbalance the loss of bone stock and stability caused by curettage, preventing impending fractures. For decades, metallic plates have been used to stabilize bones treated with curettage and bone grafting or cement filling. Although effective, metallic implants have some limitations that could be overcome using different materials [11,12,14,15,16,17,19,22,23] (Table 2).

In particular, carbon-fiber-reinforced polyetheretherketone (CFR-PEEK) implants are increasingly used in orthopedic oncology [25,30]. Our study represents the first large-sized monocentric report on using CFR-PEEK plates to stabilize bones treated with curettage due to benign and locality-aggressive tumors.

One of the main advantages of using CFR-PEEK in orthopedics is its radiotransparency [24,25,26]. This characteristic can be strategic in orthopedic oncology, particularly when combining a bone curettage with a CFR-PEEK plate. Detecting local recurrences represents a primary focus in the postoperative follow-up after a curettage [31]. Although X-rays can evidence advanced and extensive osteolytic areas suggestive of local recurrences, CT scans, and particularly MRIs, can provide more information and allow earlier detection of new neoplastic masses. However, large metallic implants such as plates produce artifacts that may alter the CT and MRI images of the nearby tissues, impeding or limiting the evaluation of potential recurrences [32]. Conversely, carbon fiber does not alter MRI scans, permitting a correct evaluation of the treated bone [24,25,26,30]. This visibility allowed us to minimize the diagnostic delay for the two cases that experienced local recurrences, discovered within only 3 and 6 months after surgery. Furthermore, the CFR-PEEK’s radiotransparency in MRI scans allowed an accurate evaluation of bone healing for those treated with bone grafts. All of them were class I according to the modified Neer classification, evidencing complete radiological healing without residual lytic areas. These findings rely on the radiological appearance of CFR-PEEK implants and their elasticity module, which is similar to native bones and allows minimal movements necessary for bone healing [24]. These properties also explain the absence of postoperative aseptic loosening and postoperative fracture rates that align with those reported in the literature for metal plates [7,8,9,10,11,12,13,14,19,22,23]. Reported fracture rates in the literature for metallic plates could be as high as 5.9% [11,15]. In contrast, in our experience, only one case (2.5%) had postoperative fractures, and none in the bone segment that received curettage and plate stabilization. Our deep infection rate, as low as 5%, also aligns with what has yet been reported for stabilizations with metal plates after bone curettage in oncologic settings [14,17,23]. These promising results and reasonable complication rates confirm the good elasticity and biological properties of carbon fiber, as well as the reliability of the implant design [33,34]. 

Our patients also had good pain resorption, functional recovery, and emotional acceptance after the treatment. None of our cases had evident pain when pressing on the surface of our carbon plates, an eventuality reported in the literature for metallic devices [17]. Most of our patients had optimal or suboptimal functional outcomes, as testified by a mean MSTS score of 27.3. In particular, our mean MSTS score for lower limbs was 26.7, slightly lower than the one obtained by Abdelrahman et al. [14], with a mean score of 28 in a significantly smaller cohort. This score is comparable to the ones published by Panchwang et al. [12], Perisano et al. [17], and Xu et al. [22], which ranged between 25.0 and 27.5. Our mean MSTS score for upper limbs was 28.0. This score was comparable to the one reported by Omlor et al. [15], who had a mean score of 28.7, but much higher than the 25.1 that Pang et al. [11] reported on their series of 49 cases. These results testify that curettage and preventive fixation with CFR-PEEK plates can lead to good functional outcomes, allowing treated patients to return to their previous activities of daily living and increasing their quality of life in the months and years to come. 

We acknowledge that our study had some limitations. The rarity of these tumors did not allow us to operate on broader populations, and the limited size of our cohort partially limited the statistical significance of some of the data associations we wanted to investigate at the beginning of our research. Another limitation is represented by the retrospective and monocentric nature of our study. The retrospective approach did not allow the complete standardization of the postoperative follow-up procedures for each patient. Our investigation’s monocentric nature relatively increased the standardization grade within our cohort but limited its size. It could also have exposed our research to unintended site-specific issues regarding clinical practice in a single institution. These limits could be overcome by performing similar evaluations on a prospective basis and with broader populations, eventually expanding our research on a multicentric scenario.

Beyond these limitations, our study provides an unprecedented evaluation of the clinical and radiographic results of curettage and prophylactic plate stabilization in cases with benign and locally aggressive bone tumors. Our results suggest that CFR-PEEK implants lead to satisfying functional performances and acceptable complication rates comparable to metal implants. Compared to this latter, however, carbon fiber allows a better view of the treated bone at the imaging examinations planned for patients’ follow-up. Our findings support using CFR-PEEK plates to stabilize bone and prevent pathological fractures in orthopedic oncology.

## 5. Conclusions

The surgical treatment of large-sized benign and locally aggressive bone lesions should be tailored depending on the necessities of each single case. The characteristics of the lesions should be carefully examined to plan the best surgical treatment. After curettage, different adjuvant treatments and fillers allow a certain grade of personalization for a large variety of conditions. CFR-PEEK plates should be considered a reliable option when stabilizing a bone after curettage and grafting since it will require postoperative bone remodeling to restore healthy bone tissue, and serial MRI evaluations will be performed to exclude local recurrences. Multicentric studies with large cohorts and longer follow-ups should be encouraged to expand our knowledge of the effectiveness of these promising implants.

## Figures and Tables

**Figure 1 jcm-14-02371-f001:**
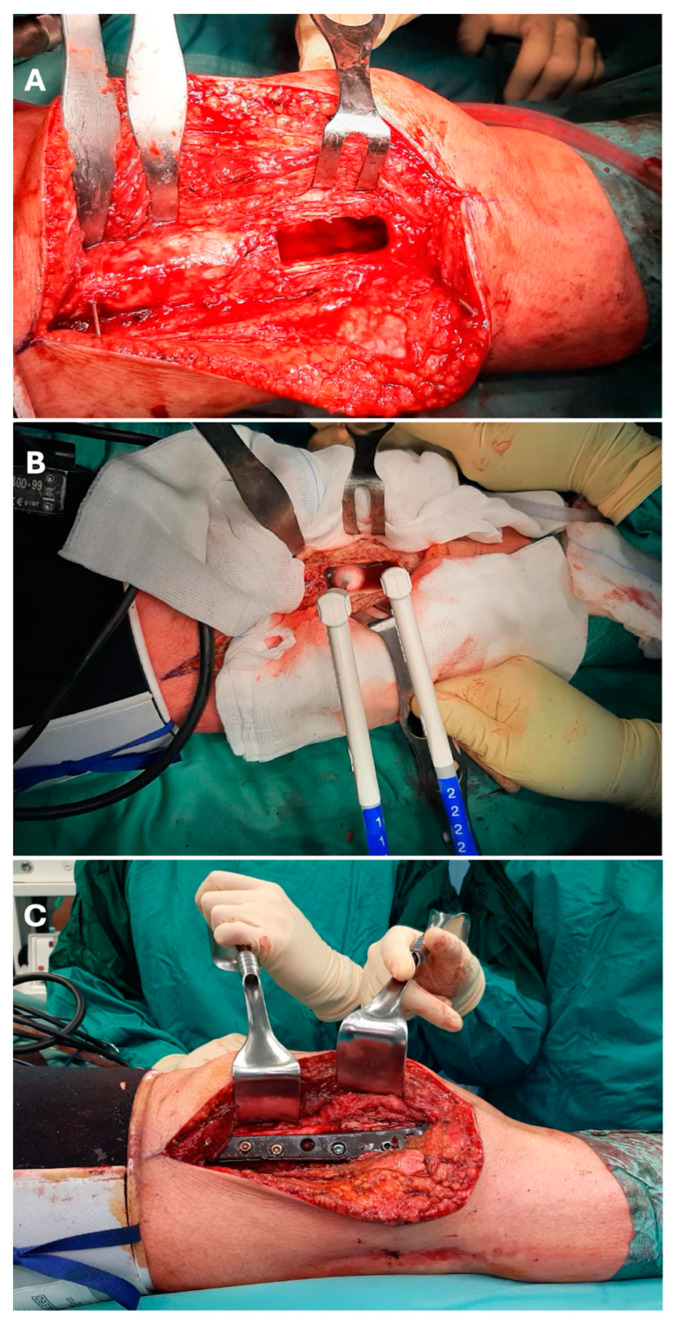
Images of a surgical intervention on an osteolytic lesion of the distal femur. A direct lateral approach to the distal femur was performed. After the curettage was completed (**A**), the resulting cavity was treated with cryotherapy using two cryoprobes (**B**). Later, the bone gap was filled with PMMA and the distal femur was stabilized using a CFR-PEEK plate and screws (**C**).

**Figure 2 jcm-14-02371-f002:**
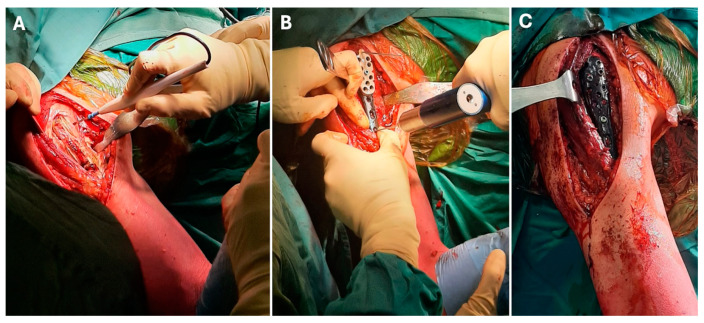
Images of a lytic lesion in the proximal humerus. Using a deltopectoral approach, the proximal humerus was exposed (**A**) for the consequential curettage and bone grafting. Later, a CFR-PEEK buttress plate is placed to cover the cortical bone (**B**) and stabilize the whole proximal humerus using compression and angular stable screws (**C**).

**Figure 3 jcm-14-02371-f003:**
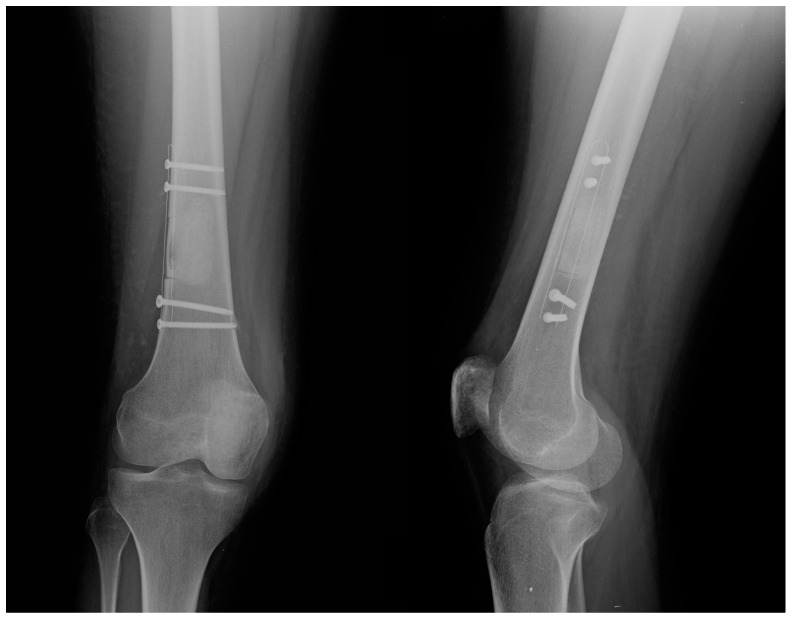
Postoperative anterior–posterior (**left**) and lateral (**right**) view of a distal femur treated with curettage, bone grafting, and stabilization using a carbon-PEEK plate.

**Figure 4 jcm-14-02371-f004:**
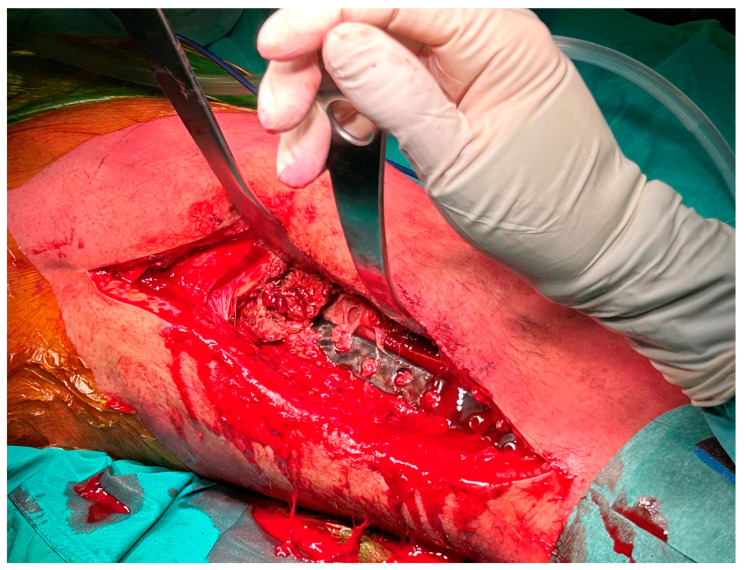
Images of a deep surgical site infection. The plate, partially covered by a membrane, was removed as well as all the screws and all the infected soft tissues.

**Figure 5 jcm-14-02371-f005:**
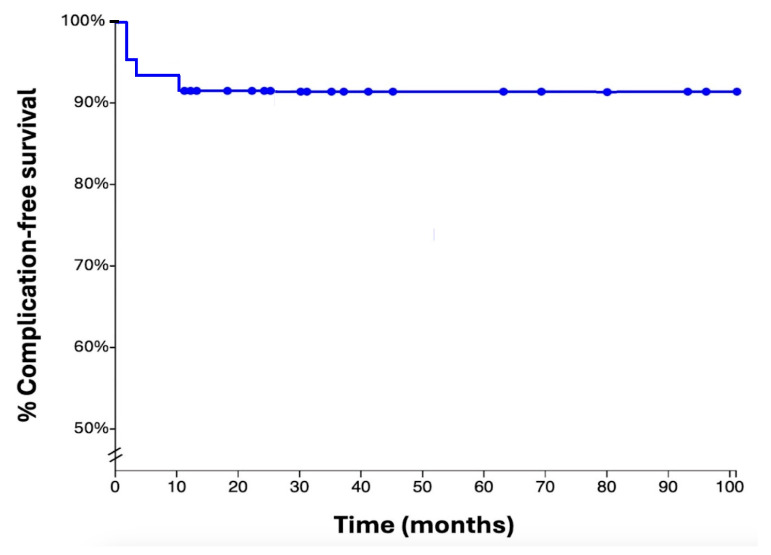
A Kaplan–Meier graph that pictures the progression of complication-free survival rates in our population. The three earliest complications (one seroma and two infections) occurred two months after surgery. One more complication (fracture) occurred 10 months after surgical treatment.

**Table 1 jcm-14-02371-t001:** A schematic summary of our cohort, divided into upper limb and lower limb lesions.

	Total	Upper Limb	Lower Limb
**NUMBER OF CASES**	**40**	**17**	**23**
**MEAN AGE**	**44.5 (14–74)**	**40.4**	**47.5**
**MEAN TUMOR SIZE**	**7.4 (3–16)**	**5.9**	**8.5**
**DIAGNOSIS**			
Atypical chondromatous tumor	17	5	12
Chondroma	11	6	5
Aneurysmal bone cyst	7	6	1
Giant cell tumor of the bone	3	0	3
Fibrous Dysplasia	1	0	1
Liposclerosing myxofibrous tumor	1	0	1
**LOCAL ADJUVANTS**			
None	20	10	10
Cryotherapy	9	2	7
Phenol	9	5	4
Alcohol	2	0	2
**FILLING**			
Allograft	30	14	16
PMMA	10	3	7
**COMPLICATIONS**	**4**	**0**	**4**
Infections	2	0	2
Seroma	1	0	1
Fracture	1	0	1
**LOCAL RECURRENCES**	**2**	**0**	**2**
**MSTS SCORE**	**27.3 (20–30)**	**28.0**	**26.7**

**Table 2 jcm-14-02371-t002:** A resume of modern literature published after 2000 on the use of curettage and bone stabilization with plates and screws. Only data derived from cases treated with curettage plate stabilization were reported for studies with heterogeneous cohorts.

Article	Year	Cases (N)	Location	Size (cm)	Material	Filling	Complications	MSTS
Abdelrahman et al. [14]	2009	6	3 Femur 3 Tibia	-	Metal	C	1 Superficial Infection	27.9
Boffano et al. [16]	2020	4	1 Prox. humerus 2 Dist. femur 1 Prox. fibia	6.6	Metal	BS	0	-
Omlor et al. [15]	2018	17	Prox. humerus	6.6	Metal	C	1 Hardware Failure 1 Fracture	28.7
Panchwang et al. [12]	2018	4	Prox. femur	8.0	Metal	BG	0	27.5
Pang et al. [11]	2023	49	Prox. humerus	6.2	Metal	BG	2 Nerve palsy 2 Fracture	25.1
Perisano et al. [17]	2016	12	Dist. femur	8.8	Metal	8 BG 4 C	2 Hardware Pain 1 Superficial Infection	-
Song et al. [23]	2024	11	Prox. femur	-	Metal	BG	1 Deep Infection	27.5
Xu et al. [22]	2021	12	3 Femur 7 Tibia 2 Foot	-	Metal	BG	1 Deformity	25.0
Zhang et al. [19]	2024	17	Prox. femur	-	Metal	BG	0	-
Ipponi et al.	2025	40	17 Prox. humerus 1 Shaft femur 19 Dist. femur 3 Prox. tibia	8.0	CFR-PEEK	30 BG 10 C	1 Deep Infection 1 Superficial Infection 1 Fracture	27.3

N = Number of cases BG = Bone Graft C = Cement.

## Data Availability

The data that support the findings of this study are available from the corresponding author upon reasonable request.

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
