# Peer review of "Fixation with Carbon Fiber Plates After Curettage in Benign and Locally Aggressive Bone Tumors: Clinical and Radiographic Outcomes"

_jcm, 2025, doi:10.3390/jcm14072371_

Round 1

Reviewer 1 Report

Comments and Suggestions for Authors

The problem of using fibroplates for fixation after curettage of benign and locally aggressive bone tumors, discussed in the article, is of considerable interest for clinical and radiological application.

Currently, metal implants are often used to close large defects. For this reason, monitoring the recovery processes by methods (radiological and radiographic), which are currently widely used in clinical practice, may not fully reflect the real picture of recovery. In addition, strong elastic materials for closing bone defects, filled with cements or bone grafts, do not lead to increased pain during examination of the damaged area. The authors of the article described this problem in sufficient detail. In this regard, the study presented in the article, in my opinion, is original. There are few cases of using fibroplates in comparison with metal implants in the literature. In this paper, clinical cases using fibrous materials are analyzed and systematized.

The authors analyzed clinical cases observed over a fairly long period (from July 2016 to February 2024), which allows them to make fairly correct conclusions. However, a significant drawback of this article is the inaccuracy in the name of the material used. Carbon fibers and PEEK (polyetheretherketone) are completely different in their chemical nature, so in all sections of the article (Abstract, Keywords, Introduction, Materials and Methods, Results, Discussion, conclusions) it is necessary to accurately indicate which materials were used.

In addition, it is necessary to indicate the manufacturer of these materials. Without these corrections, the information in the entire article may be misinterpreted. As for stylistic shortcomings, it should be noted that when using abbreviations, they must be expanded or put in a separate list.

In different areas of application, the same abbreviation can mean completely different things. Stylistic shortcomings also include careless formatting of tables. The design should be uniform for better perception of the information presented.

Author Response

Dear Reviewer ,

Thank you for your help and your suggestions to increase the quality of our paper.

These are our replies to your suggestions:

  • The authors analyzed clinical cases observed over a fairly long period (from July 2016 to February 2024), which allows them to make fairly correct conclusions. However, a significant drawback of this article is the inaccuracy in the name of the material used. Carbon fibers and PEEK (polyetheretherketone) are completely different in their chemical nature, so in all sections of the article (Abstract, Keywords, Introduction, Materials and Methods, Results, Discussion, conclusions) it is necessary to accurately indicate which materials were used.
    We are conscious and agree that carbon fibers and PEEK are completely different materials. In fact, as already reported in lines 63-64, CFR-PEEK plates are made of "Carbon-fiber-reinforced polyetheretherketone (CFR-PEEK)." We changed the text to replace the use of carbon fiber alone in several sentences with a more correct CFR-PEEK to reduce the risk of misunderstandings. We also added some information about the composition of the used plates (lines 115-117). 

  • In addition, it is necessary to indicate the manufacturer of these materials. Without these corrections, the information in the entire article may be misinterpreted.
    Please note that the implants' brand and manufacturer of these materials had already been reported (Piccolo Composite™ Plating System, CarboFix Orthopedics®, Herzliya, Israel) in what is now line 114.

  • As for stylistic shortcomings, it should be noted that when using abbreviations, they must be expanded or put in a separate list.
    In different areas of application, the same abbreviation can mean completely different things. 
    We agree that explaining abbreviations is necessary to allow readers to comprehend them better. We could find two abbreviations not previously described (MSTS for musculoskeletal society scoring system in line 147 and GCT for giant cell tumor of the bone in line 163). As far as we could see, no other explanation for an abbreviation was missing. 

Please, find the changes written in red in the revised manuscript.

Best regards

Reviewer 2 Report

Comments and Suggestions for Authors
  1. Title and Abstract

Strengths:

The title is clear and descriptive, indicating the study's focus on carbon fiber plates in orthopedic oncology. The abstract provides a concise summary of the study's background, methods, results, and conclusions.

Key findings, such as the MSTS scores and complication rates, are highlighted effectively.

Suggestions for Improvement:

The abstract could briefly mention the statistical significance of the findings to emphasize their robustness.

  1. Introduction

Strengths:

The introduction provides a background on benign bone tumors and their treatment options. It clearly outlines the limitations of metallic plates and introduces carbon fiber-reinforced polyetheretherketone (CFR-PEEK) as an alternative.

Suggestions for Improvement:

The introduction could benefit from a more detailed discussion of previous studies on CFR-PEEK plates to establish a stronger foundation for the research. The rationale for the study is well-articulated but, more can be discussed for the argumentation of the technique incl. new papers as the systematic review of Rijs et al. Journal of Orthopaedics and Traumatology https://doi.org/10.1186/s10195-023-00724-4  (2023)

Including a brief comparison to metallic plates in the abstract might enhance its appeal to readers unfamiliar with carbon fiber plates.

  1. Materials and Methods

Strengths:

The study design (retrospective single-center) and inclusion/exclusion criteria are clearly defined. Surgical techniques are described in detail, including imaging protocols, curettage methods, and fixation procedures. Ethical compliance with the Declaration of Helsinki is noted.

Suggestions for Improvement:

More information on patient selection criteria (e.g., how cases with impending fractures were identified) could strengthen reproducibility. Consider elaborating on why certain local adjuvants (e.g., cryotherapy vs. phenol) were chosen for specific cases.

  1. Results

Strengths:

The results are well-organized and provide detailed data on demographics, tumor characteristics, treatments, complications, recurrences, and functional outcomes.

Visual aids (e.g., figures showing surgical interventions and postoperative X-rays) enhance clarity.

Suggestions for Improvement:

The Kaplan-Meier graph for complication-free survival rates is mentioned but not fully explained. Adding interpretation would help readers understand its significance.

The recurrence cases could be discussed more thoroughly to identify potential risk factors or patterns.

Statistical analyses could be expanded upon to provide more robust evidence for the conclusions drawn.

Additional context on patient-reported outcomes or quality-of-life measures post-surgery would enhance clinical relevance.

  1. Discussion

Strengths:

The discussion highlights the advantages of CFR-PEEK plates, including radio-transparency and mechanical properties. Postoperative functionality scores (MSTS) are contextualized within the broader orthopedic oncology literature. The article provides valuable insights into an emerging technology in orthopedic oncology.

Suggestions for Improvement:

A direct comparison between CFR-PEEK plates and traditional metallic plates in terms of cost-effectiveness, long-term outcomes, or patient satisfaction would add depth.

Addressing limitations (e.g., small sample size, single-center design) explicitly would strengthen the study's credibility.

Some papers – incl. from current Journal can be quoted and discussed…

Shain Patel - Carbon Fiber Implants in Orthopaedic Oncology  Article in Journal of Clinical Medicine (JCM) · August 2022

Marilee J. Clunk  - A PEEK into carbon fiber: A practical guide for high performance composite polymeric implants for orthopaedic oncology Article in Journal of Orthopaedics 2023 https://doi.org/10.1016/j.jor.2023.09.011

  1. Conclusion

Strengths:

The conclusion succinctly summarizes the study's findings and emphasizes the potential of CFR-PEEK plates in personalized surgical approaches.

Suggestions for Improvement:

Including a call for larger multicenter studies or randomized controlled trials would highlight future research directions.

  1. Final Recommendation

This article is a strong contribution to orthopedic oncology literature. With minor revisions to expand on comparisons with metallic plates, statistical analyses, and limitations, it could become even more impactful.

Author Response

Dear Reviewer ,

Thank you for your help and your suggestions to increase the quality of our paper.

These are our replies to your suggestions:

  • The abstract could briefly mention the statistical significance of the findings to emphasize their robustness.
    Statistical analyses could be expanded upon to provide more robust evidence for the conclusions drawn.
    Unfortunately, the size of our cohort and the nature of our study (including having a single cohort without a control group) did not allow for statistical analysis, including contingency tables or linear correlations. Therefore, our results only consist of the raw mean values obtained, including scores, complication rates, and local recurrences.

  • The introduction could benefit from a more detailed discussion of previous studies on CFR-PEEK plates to establish a stronger foundation for the research. The rationale for the study is well-articulated but, more can be discussed for the argumentation of the technique incl. new papers as the systematic review of Rijs et al. Journal of Orthopaedics and Traumatology https://doi.org/10.1186/s10195-023-00724-4  (2023)
    Including a brief comparison to metallic plates in the abstract might enhance its appeal to readers unfamiliar with carbon fiber plates.
    To follow your suggestions, we tried to increase the quality of our introduction by further opposing the already-reported characteristics of metal plates (lines 57-62) with a more detailed characterization of CFR-PEEK implants (lines 64-75), relying on the systematic review by Rijs et al. as suggested.

  • More information on patient selection criteria (e.g., how cases with impending fractures were identified) could strengthen reproducibility. Consider elaborating on why certain local adjuvants (e.g., cryotherapy vs. phenol) were chosen for specific cases.
    Impending fractures were identified with the Mirels' criteria (line 92). Local adjuvants were used mainly for selected cases with massive scalloping or a known diagnosis of locally aggressive tumor (line 103). We could not provide further rationale on the adjuvant type since the choice of one over the other relied more on the preference of the single surgeon rather than on a shared univocal approach.

  • The Kaplan-Meier graph for complication-free survival rates is mentioned but not fully explained. Adding interpretation would help readers understand its significance.
    As suggested, we implemented the description of our KM graph in lines 192-193

  • The recurrence cases could be discussed more thoroughly to identify potential risk factors or patterns.
    As suggested, we provided more data about both patients who developed local recurrences. We reported their lesions' size (one above and one below the average for our cohort; lines 195-196), and we described their recurrent lesions' diagnosis and treatment (lines 199-214)

  • Additional context on patient-reported outcomes or quality-of-life measures post-surgery would enhance clinical relevance.
    We acknowledge that scores like QOLS could have added further relevance to our study, but we do not routinely perform this scale, so we could not include it in our retrospective study. However, please consider that the MSTS scoring system not only considers functional outcomes but also patients' emotional acceptance and satisfaction (accounting for 1 to 5 points). We emphasized it in lines 311 and 312.

  • A direct comparison between CFR-PEEK plates and traditional metallic plates in terms of cost-effectiveness, long-term outcomes, or patient satisfaction would add depth.
    We agree that comparisons between CFR-PEEK and traditional metallic plates would be intresting. Although the nature of our study (single cohort and not a case-control study) does not allow a direct comparison within our population, we tried to follow your suggestion by further comparing our outcomes with the ones obtained by others in similar conditions using metal plates (lines 304-306, 315-321). 

  • Addressing limitations (e.g., small sample size, single-center design) explicitly would strengthen the study's credibility.
    We agree and explicitly included these among our study limitations (lines 326 and 329-335).

  • Some papers – incl. from current Journal can be quoted and discussed (Shain Patel - Carbon Fiber Implants in Orthopaedic Oncology  Article in Journal of Clinical Medicine (JCM) · August 2022; Marilee J. Clunk  - A PEEK into carbon fiber: A practical guide for high performance composite polymeric implants for orthopaedic oncology Article in Journal of Orthopaedics 2023 https://doi.org/10.1016/j.jor.2023.09.011)
    We added these two interesting articles to our discussion in line 310.

Please, find the changes written in red in the revised manuscript.

Best regards